# A Hybrid Self-Supervised Visual Representation Learning Method for histopathological image classification

## Abstract

Extracting visual representations is a crucial challenge in the domain of computational histopathology. Considering the powerful strength of deep learning algorithms and the dearth of annotated samples, self-supervised learning presents itself as a compelling strategy to extract effective visual representations from unlabeled histopathology images. Although some self-supervised learning methods have been specifically proposed for histopathology image classification, most of them have certain drawbacks that may affect the functionality or representation capacity. In this work, we propose Masked Mamba, a novel self-supervised visual representation learning method tailored for histopathology images that can adequately extract local-global features. The proposed method consists of two stages: local perception positional encoding (LPPE) and directional Mamba vision backbone (DM). In addition, we use masked autoencoder (MAE) pretraining to unleashing directional Mamba vision backbone's potential. Masked Mamba makes good use of domain-specific knowledge and requires no side information, which means good rationality and versatility. Experimental results demonstrate the effectiveness and robustness of masked Mamba on common histopathology classification tasks. Furthermore, ablation studies prove that the local perception positional encoding and directional Mamba vision backbone in masked Mamba can complement and enhance each other.

## 1 Introduction

Histopathology plays an important role in clinical medicine. It can reveal the morphology of pathological cells and tissues under a microscope and provide key information for disease diagnosis Srinidhi et al. (2021). With the histopathological slides have been digitized as histopathological images, computer-aided histopathological image analysis methods have been widely proposed Mobadersany et al. (2018). In early researches of histopathological image classification, the features of histopathology are manually designed and extracted via traditional feature extraction methods. However, these handcrafted features are very subjective and less representation capacity Madabhushi & Lee (2016). Recently, deep learning-based methods have shown strong representation capabilities LeCun et al. (2015), but such methods rely on large amounts of labeled data to learn visual representations. Large-scale labeled datasets are expensive and time-consuming for histopathological image data. Therefore, researchers utilize pre-trained deep models, e.g. ImageNet Deng et al. (2009) pre-trained convolutional neural Network (CNN), to extract visual representations histopathological images Senousy et al. (2021). However, this strategy ignores data distribution differences and task biases, which will lead to inappropriate or suboptimal visual representations.

Therefore, self-supervised learning (SSL) Azizi et al. (2021) is one of the feasible solutions in histopathological image classification. SSL can only use unlabeled data to adapt deep models. The deep model can be easily trained to capture the features in a supervised learning manner. For natural images, self-supervised learning methods based on contrastive learning (CL) Zhang et al. (2022b) and masked image model (MIM) Chen et al. (2024) have achieved amazing results and shrunk the performance gap with supervised methods on downstream tasks Jing & Tian (2020). However, there are three aspects that could be further enhanced. First, in the histopathological image classification task, rotation and shift operations should not alter the final result of the model. In

other words, we expect translation invariance Kayhan & Gemert (2020) in histopathological image classification. However, the absolute position encoding, initially designed to leverage the order of tokens, damages such invariance because it adds unique positional encoding to each patch Chu et al.. Second, the cropped histopathological image patches are typically large enough to capture both cell-level structures (e.g., cellular microenvironment) and tissue-level contexts (e.g., tumor microenvironment). Thus, both local and global features are advantageous for histopathological image analysis and should be extracted. Convolutional Neural Network (CNN) Lerousseau et al. (2020) have a strong capacity for learning low-level texture content features (local features). But the learning of global context features is often constrained by the receptive field of CNNs. Transformer-based algorithms Stegmüller et al. (2023) can capture long-distance dependencies(global features) through self-attention mechanisms. But high computational costs and reliance on large-scale data limit their performance in histopathological image classification. In Mamba-based algorithms Gu & Dao (2023), the State Space Model (SSM) Hamilton (1994) is used to effectively capture the local and global features. But Mamba is suitable for tasks with long sequences and autoregressive features Yu & Wang (2024). These advantages cannot be exploited in the histopathology classification task Yue & Li (2024). Third, the task of histopathology classification focuses on transferability. Compared with contrastive learning algorithms that rely too much on data comparison, the masked image model can not only save computational cost, but also be applied to medical images without data augmentation Qi et al. (2023); Zhou et al. (2023a).

To this end, we propose Masked Mamba, a novel hybrid self-supervised visual representation learning method tailored for H&E-stained histopathological images. Our Masked Mamba employs two stages for histopathological image classification. One is the local perception positional encoding (LPPE), and the other is the directional Mamba vision backbone (DM). And we use masked autoencoder (MAE) He et al. (2022) pretraining to unleashing our directional Mamba vision backbone's potential. The major contributions of our work are summarized as follows.

- We propose the LPPE can help capture both local and global structure information within the features and promote the representation ability of the network.
- We construct a hybrid architecture (DM) for histopathological image classification. It replaces the causal convolution of Mamba with depthwise separable convolution and standard convolution, which enables more stable network training and also helps build a powerful feature extractor with fine local structure and global context. Not only that, such a structure is more suitable for MAE than the original Mamba Liu & Yi (2024).
- To the best of our knowledge, this is the first hubrid Mamba-based unsupervised feature extractor carried out on the public histopathological image datasets. We use the MAE to motivate the potential of our DM.
- The efficacy of Masked Mamba is empirically substantiated through rigorous testing on four publicly available histopathological datasets. The empirical evidence showcases the superior performance of our algorithm when juxtaposed with existing state-of-the-art (SOTA) methodologies, thereby marking a significant leap forward in the domain of histopathological image classification.

## 2 RELATED WORK

**Mamba Vision.** With the advent of the Mamba model in the natural language processing (NLP) Gu & Dao (2023), some studies have used it for computer vision tasks. Specifically, Vision Mamba (ViM) Zhu et al. (2024) proposed the use of a bidirectional SSM formulation, which sets tokens in both forward and backward directions to obtain information. VMamba Liu et al. (2024) introduced a Cross-Scan Module (CSM) that employs a four-way selective scanning method (i.e., from upper-left to lower-right and vice versa), facilitating 1D selective scanning. EfficientVMamba Pei et al. (2024) proposed an atrous selective scanning method combined with skip sampling, effectively extracting global spatial dependencies. LocalMamba Huang et al. (2024) adopts an approach similar to Swin Transformer Liu et al. (2021) to divide the image into different Windows, effectively capturing local dependencies while maintaining a global perspective. Even though Mamba-based model was recently introduced to address the quadratic complexity of the attention mechanism in computer vision. But its the performance is often underwhelming when compared with the CNN-based and Transformer-based models in histopathological image classification. The reason for this

phenomenon is that Mamba-based model is ideally suited for tasks with long-sequence and autoregressive characteristics. However, the histopathology image classification task does not align with either characteristic.

**Self-supervised learning in Histopathological Image Classification.** Recently, SSL, as an unsupervised learning paradigm, has achieved extraordinary performance in the field of pathological image analysis. These techniques can be broadly categorized into CL and MIM-based methods. SSL is typically divided into two main categories: contrastive and generative. In the context of medical imaging, current applications of CL include the following: Li et al. (2021a) propose the use of self-supervised contrastive learning to extract robust representations for Multiple Instance Learning (MIL) Deng et al. (2024). Ciga et al. (2022) introduce a contrastive self-supervised learning method applied to large-scale pathology datasets from multiple organs with varying types of stains and resolutions. Huang et al. (2021) extract patch features from Whole Slide Images (WSIs) through self-supervised learning and adaptively aggregate these features based on their spatial information and inter-patch correlation using the Transformer architecture. Li et al. (2021b) emphasize that patch-wise spatial proximity is a significant characteristic of WSIs. Abbet et al. (2020) propose a self-supervised learning method that jointly learns a representation of tissue regions and a clustering metric to uncover their underlying patterns. Vu et al. (2023) present a handcrafted framework based on deep Convolutional Neural Networks (CNNs) for classifying different cancer subtypes. A typical algorithm based on MIM is the MAE. For instance, Zhou et al. (2023b) investigate a MAE-based self-pretraining paradigm for the classification of diseases in chest X-rays, multi-organ segmentation in abdominal CT scans, and the segmentation of brain tumors in MRI. Zhang et al. (2022a) propose a family of MAE for electrocardiographs, which includes three customized masking modes: the masked time autoencoder, the masked lead autoencoder, and the masked lead and time autoencoder. Chen et al. (2023) study the strategies of how masked image modeling can enhance performance from the perspectives of 3D medical image segmentation. Dai et al. (2023) propose a MAE integrated with the Swin Transformer and note its suitability for smaller medical datasets. Quan et al. (2024) propose a global contrast-masked autoencoder capable of capturing both local and global features of pathological images.

## 3 METHOD

### 3.1 PRELIMINARIES

#### 3.1.1 STATE SPACE MODELS

SSMs are a general family of sequence models used in deep learning, influenced by systems capable of continuously mapping one-dimensional sequences. These models transform input sequence $x(t) \in \mathbb{R}^{L \times D}$ into output sequence $y(t) \in \mathbb{R}^{L \times D}$ by utilizing a learnable latent state $h(t) \in \mathbb{R}^{N \times D}$ that is not directly observable. The mapping process could be denoted as:

$$
\begin{aligned}
h'(t) &= \mathbf{A}h(t) + \mathbf{B}x(t) \\
y(t) &= \mathbf{C}h(t)
\end{aligned}
\tag{1}
$$

where $\mathbf{A} \in \mathbb{R}^{N \times N}$ represents the state matrix, $\mathbf{B} \in \mathbb{R}^{N \times 1}$ and $\mathbf{C} \in \mathbb{R}^{N \times 1}$ denote the projection parameters. The Eq. 1 is transformed into a discrete function to achieve more efficient computation. Therefore, SSMs are discretized using the zero-order hold rule at a given sampling time scale $\Delta \in \mathbb{R}^D$ as follows:

$$
\begin{aligned}
\overline{\mathbf{A}} &= e^{\Delta \mathbf{A}} \\
\overline{\mathbf{B}} &= (e^{\Delta \mathbf{A}} - \mathbf{I})\mathbf{A}^{-1}\mathbf{B} \\
\overline{\mathbf{C}} &= \mathbf{C} \\
\overline{\mathbf{B}} &\approx (\Delta \mathbf{A})(\Delta \mathbf{A})^{-1}\mathbf{A}\mathbf{B} = \Delta \mathbf{B} \\
h(t) &= \overline{\mathbf{A}}h(t-1) + \overline{\mathbf{B}}x(t) \\
y(t) &= \mathbf{C}h(t)
\end{aligned}
\tag{2}
$$

where $\overline{\mathbf{A}} \in \mathbb{R}^{N \times N}$, $\overline{\mathbf{B}} \in \mathbb{R}^{D \times N}$ and $\overline{\mathbf{C}} \in \mathbb{R}^{D \times N}$.

### 3.1.2 SELECTIVE STATE SPACE MODELS

Selective State Space Models (S6) enhance the information processing capabilities across sequences by diffusing the discretization process through a selection mechanism in Mamba.

$$\begin{aligned}
\overline{\mathbf{B}} &= s_{\mathbf{B}}(x) \\
\overline{\mathbf{C}} &= s_{\mathbf{C}}(x) \\
\Delta &= \tau_{\mathbf{A}}(Parameter + s_{\mathbf{A}}(x))
\end{aligned} \tag{3}$$

where $s_{\mathbf{B}}(x)$ and $s_{\mathbf{C}}(x)$ are linear functions that project input x into an N- dimensional space, $s_{\mathbf{A}}(x)$ is a function that adjusts selectively based on the input, which can be either linear or nonlinear. $\tau_{\mathbf{A}}$ is a scaling factor, $Parameter$ represents the base parameters. On the basis of the above, VMamba proposed the 2D Selective Scan (SS2D) for visual tasks, which maintains the integrity of 2D image structures by scanning four directed feature sequences. Each sequence is processed independently within an S6 block and then being combined to form a comprehensive 2D feature map.

### 3.1.3 MASKED IMAGE MODELING

MIM approaches are generally characterized by a two-pronged approach: pretraining and finetuning for downstream tasks. The objective of the pretraining, often referred to as the surrogate task, entails the obfuscation of a subset of image patches and the subsequent endeavor to regenerate these masked patches from within the confines of the original image. This surrogate task within the MIM framework can be shown as follows:

$$\mathcal{L}_{MIM} = f_{mask}(x) \rightarrow \tilde{x} \tag{4}$$

where $x$ and $\tilde{x}$ denote the original and the regenerated images, respectively. The discrepancy between $x$ and $\tilde{x}$ is typically quantified using the mean squared error (MSE) computed on a per-pixel basis, serving as the pretraining loss function, which is articulated as:

$$\mathcal{L}_{MSE} = \sqrt{\frac{1}{N} \sum_{i=1}^{N} (x_i - \tilde{x}_i)^2} \tag{5}$$

where $N$ represents the total number of pixels. Upon the completion of the pretraining , the derived feature representations are then transposable to a spectrum of supervised learning tasks in downstream applications.

### 3.2 MASKED MAMBA MODEL: OVERVIEW

The detailed procedure of our Masked Mamba pretraining model is delineated in Figure 1. It consists three stages. Firstly, the LPPE is proposed to capture both local and global structure information within the features. Some researchers have found that the performance of Mamba model degrades when pretraining Mamba by MEA. This is because MAE is not compatible with bidirectional state space blocks in Mamba. Therefore, we redesign the original Bi-Mamba Zhu et al. (2024) architecture to better accommodate tasks associated with MAE. Finally, we use randomly sample the masking ratio (0.75) to unleashing directional Mamba vision backbone's potential. Masked Mamba makes good use of domain-specific knowledge and requires no side information, which means good rationality and versatility. The implementation of the Masked Mamba pretraining is straightforward and can be abstractly represented by Equ. 6:

$$X \rightarrow Masked(X) \rightarrow X_m \rightarrow MaskedMamba \rightarrow H \rightarrow Decoder \rightarrow \hat{X} \tag{6}$$

Firstly, the histopathological image pixels $X \in \mathbb{R}^{H \times W \times 3}$, where $H$ and $W$ represent the height and width of the input image, respectively. Then, we randomly sample the masking ratio ($m_r = 0.75$), and mask out $m_r \cdot (H \cdot W)$ tokens, replacing them with a learnable mask token ($X_m$). Subsequently, we transform the class id into a learnable label embedding, denoted as $cls$. The masked image $X_m$ is utilized as the input for the encoder, which generates multi-scale latent representations denoted by $H$. Finally, the decoder receives the representation $H$ and produces a reconstructed image $\hat{X}$. During the pretraining, we employ the pixel-wise MSE, as defined in Equ. 5, as the loss function. Unlike the MAE method, we design a multi-scale encoder structure that can effectively capture both short-range and long-range information. The decoder employs a simple Masked Mamba block to reconstruct the pixels of the original image from the encoded visible patches and masked tokens.

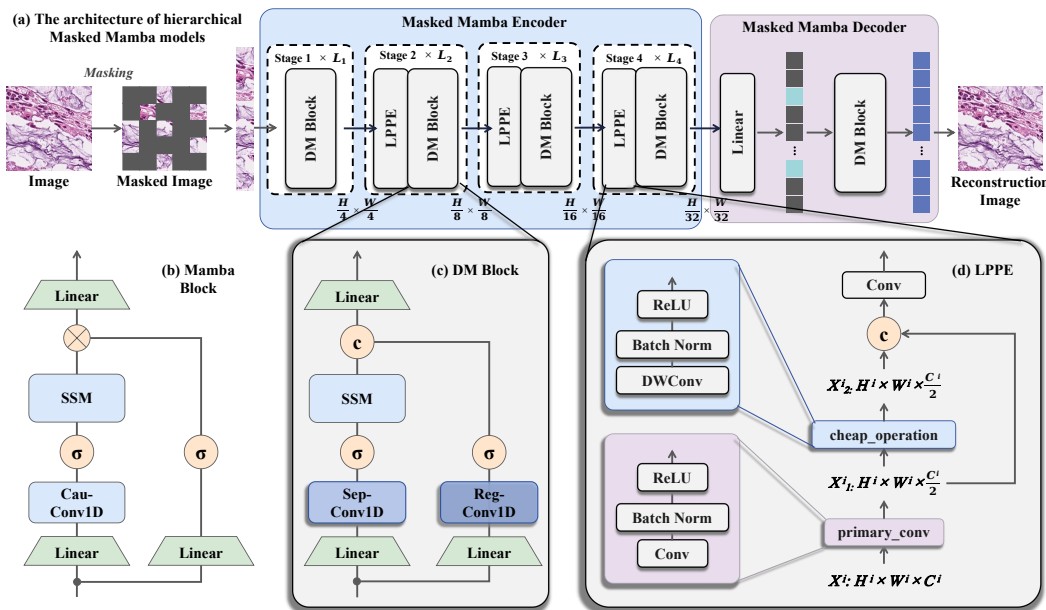

Figure 1: The Masked Mamba's framework. (a) The overall structure of Masked Mamba. (b) The original Mamba. (c) Our DM Block (d) Our LPPE structure.

## 3.3 MASKED MAMBA: ARCHITECTURE

### 3.3.1 THE LOCAL PERCEPTION POSITIONAL ENCODING (LPPE)

Due to the staining and sectioning of histopathological images, the consistency of the images may be affected, and the highly similar cells make the classification task more difficult than general classification tasks. Therefore, we need to capture both local and global structural information in the features to promote the representation ability of the network. On this basis, the absolute positional encoding used in previous transformers and Mamba, initially designed to leverage the order of tokens, damages the translation-invariance because it adds unique positional encoding to each patch. Therefore, we design a novel positional encoding strategy (LPPE), which differs from the linear operation of traditional ViT models, It uses a combination method of convolution operation, linear and residual connection to extract local information. At the same time, it differs from Swin Transformer in the way of Patch Merging. As shown in Figure 2, we compare the Patch Merging in Swin Transformer with our LPPE. Patch Merging reduces the resolution of feature maps by merging neighboring patches using downsampling techniques. However, the resolution in pathological images is often affected by staining and sectioning, and downsampling can exacerbate the above effects. Our design of LPPE is inspired by the Local Perception Unit (LPU) in CMT Guo et al. (2022). Based on LPU, we introduce residual networks. The DWConv is used to extract local information with negligible extra computational cost. The motivation for inserting shortcut is similar to that of classic residual networks, which can promote the propagation ability of gradient across layers.

Specifically, $X^i \in \mathbb{R}^{H^i \times W^i \times C^i}$ is the resolution of the input of current stage, $C^i$ indicates the dimension of features, and $H^i$ and $W^i$ are the height and width of features, respectively. Then, intrinsic feature maps with $\frac{C^i}{2}$ channel dimensions are generated through a primary convolution $X_1^i \in \mathbb{R}^{H^i \times W^i \times \frac{C^i}{2}}$. A Depthwise Convolution (DWConv) operation is applied to each intrinsic feature in $X_1^i$ to generate similar features with $\frac{C^i}{2}$ channel dimensions. Finally, the output with the same dimension as the input is obtained through feature merging on the channel dimension. To enhance the entire encoder's ability to capture multi-scale features, we have added a convolutional layer to enhance discriminative features and contribute to further performance improvements. LPPE

Figure 2: Positional Encoding strategy. (a) The patch merging of Swin Transformer. (b) Our LPPE.

can be defined as:

$$
\begin{aligned}
X^i &= \text{ReLU}(\text{BN}(\text{Conv}(X^i))) \\
X_{Final} &= \text{Conv}(\text{ReLU}(\text{BN}(\text{DWConv}(X_1^i)))) + X^i
\end{aligned}
\tag{7}
$$

where BN is the batch norm layer, Conv signifies the regular convolution, DWConv is the depthwise convolution.

### 3.3.2 THE DIRECTIONAL MAMBA VISION BACKBONE

we redesigned the original Mamba mixer As depicted in Figure 1 (c), we redesign the original Mamba (as shown in Figure 1 (b)) to make it more suitable for vision tasks. First, we replace the causal convolution (Cau-Conv1D) with depthwise separable convolution (Sep-Conv1D). Given that samples in pathology images (such as different types of cancer cells) may be visually highly similar, the causal convolution only permits unidirectional token mixing, which hinders the potential of non-autoregressive image generation. In contrast, the standard convolution enables tokens to interact bidirectionally across all positions in the input sequence, effectively capturing the global context. In addition, we added a symmetric branch without SSM, consisting of a regular convolution (Reg-Conv1D) and SiLU activation, to compensate for any content lost due to the sequential constraints of SSMs. We then concatenate the output of both branches and project it via a final linear layer. This combination ensures that the final feature representation incorporates both the sequential and spatial information, leveraging the strengths of both branches. We note that the output of each branch is projected into an embedding space with size $C/2$ (i.e., half the size of the original embedding dimension) to maintain a similar number of parameters to the original block design. Given an input $X_{in}$, the output $X_{out}$ of the DM Block can be computed as follows:

$$
\begin{aligned}
X_1 &= \text{Scan}(\sigma(\text{Sep-Conv1D}(\text{Linear}(C, \tfrac{C}{2})(X_{in})))) \\
X_2 &= \sigma(\text{Reg-Conv1D}(\text{Linear}(C, \tfrac{C}{2})(X_{in}))) \\
X_{out} &= \text{Linear}(\tfrac{C}{2}, C)(\text{Concat}(X_1, X_2))
\end{aligned}
\tag{8}
$$

where $\text{Linear}(C_{in}, C_{out})(\cdot)$ denotes a linear layer with $C_{in}$ and $C_{out}$ as input and output embedding dimensions, Scan is the selective scan operation as in Gu & Dao (2023) and $\sigma$ is the activation function. In addition, Sep-Conv1D is the depthwise separable convolution, Reg-Conv1D signifies the 1-dimensional regular convolutional operation, and Concat indicates the concatenation operation. In general, our proposed modification leads to richer feature representations, better generalization, and improved performance on histopathology image classification tasks.

### 3.3.3 MASKED MAMBA TRANSFER

After Masked Mamba pretraining, the pre-trained Masked Mamba Encoder is transferred to downstream task to evaluate the effectiveness of our Masked Mamba approach. Therefore, the general pipeline of our Masked Mamba transfer is:

$$
x \xrightarrow{\text{Masked Mamba Encoder}} H \xrightarrow{\text{Classification Head}} \hat{y}
\tag{9}
$$

where the $\hat{y}$ stands for the predicted target (image-wise labels for classification). Following He et al. (2022), the classification task head consists of one linear layer appended to the Masked Mamba Encoder, which receives the latent representations $H$ and predicts classification labels. The binary cross entropy (BCE) loss is used for classification:

$$
L_{BCE} = -\frac{1}{N} \sum_{n=1}^{N} [y_n log(\hat{y}_n) + (1 - y_n) log(1 - \hat{y}_n)]
\tag{10}
$$

where $y_n$ and $\hat{y}_n$ represents ground-truth and predicted label for the $n$th input image, respectively.

Table 1: The 4 Pathology image dataset details.

| Dataset | Label Nums | Image Pixel | Image Nums | Organ/Tissue | Field |
|---|---|---|---|---|---|
| LaC Borkowski et al. (2019) | 5 | $768 \times 768$ | 25000 | Colon and Rectum | Histopathology |
| NCT Kather et al. (2019) | 9 | $224 \times 224$ | 100000 | Colon and Rectum | Histopathology |
| PBC Acevedo et al. (2020) | 8 | $360 \times 363$ | 17092 | Blood | Cytopathology |
| TCGA COAD Kirk et al. (2016) | 2 | $224 \times 224$ | 192312 | Colon and Rectum | Histopathology |

## 4 EXPERIMENTAL RESULTS

### 4.1 DATASETS AND EXPERIMENTS IMPLEMENTATION

Our experiments contain 4 publicly available pathology image datasets, which include the Lung and Colon Cancer (LaC) Borkowski et al. (2019), NCT-CRC-HE-100K (NCT) Kather et al. (2019), Peripheral Blood Cell (PBC) Acevedo et al. (2020), and The Cancer Genome Atlas Colon Adenocarcinoma (TCGA COAD) Kirk et al. (2016). The 4 datasets are listed in detail in Table 1 and A.2.

**LaC** contains color 25,000 images with 5 classes (colon adenocarcinoma, benign colon tissue, lung adenocarcinoma, lung squamous cell carcinoma and benign lung tissue) of 5,000 images each. All images are $768 \times 768$ pixels in size and are in jpeg file format.

**NCT** is a pathology image dataset designed for image classification, comprising 100,000 hematoxylin and eosin stained histological images of human colorectal cancer and healthy tissues extracted from 86 patients. The dataset includes images of nine different tissue types, which have been color normalized using the Macenko method to effectively reduce color variations from different slides. It contains colorectal cancer and normal histology images, which is composed of $10,407$ adipose (ADI), $10,566$ background (BACK), $11,512$ debris (DEB), $11,557$ ymphocytes (LYM), $8896$ mucus (MUC), $13,536$ smooth muscle (MUS), $8,763$ normal colon mucosa (NORM), $10,446$ cancer-associated stroma (STR) and $14,317$ colorectal adenocarcinoma epithelium (TUM).

**PBC** contains a total of 17,092 images of individual normal cells, which were acquired using the analyzer CellaVision DM96 in the Core Laboratory at the Hospital Clinic of Barcelona. The dataset is organized in the following eight groups: neutrophils, eosinophils, basophils, lymphocytes, monocytes, immature granulocytes (promyelocytes, myelocytes, and metamyelocytes), erythroblasts and platelets or thrombocytes. The size of the images is $360 \times 363$ pixels, in format jpg, and they were annotated by expert clinical pathologists.

**TCGA COAD** contains 192312 unique image patches derived from histological images of colorectal cancer and gastric cancer patients in the TCGA cohort. The dataset encompasses two categories: microsatellite stable (MSS) and microsatellite unstable or highly mutated (MSI). It has been utilized for the automatic detection of tumors. The slice size is $224 \times 224$ pixels, the color is normalized by the Macenko method, and the format is JPG. The TCGA-COAD data collection is part of a larger effort to build a research community focused on connecting cancer phenotypes to genotypes by providing clinical images matched to subjects from the TCGA. Clinical, genetic, and pathological data resides in the Genomic Data Commons Data Portal while the radiological data is stored on The Cancer Imaging Archive.

We employ two commonly used metrics: accuracy(Acc) and F1-score(F1) to evaluate our proposed framework quantitatively. In this work, we use graphic card NVIDIA RTX A5000 (24GB) for the training and testing. The PyTorch version used for the implementation is 2.10.0, the Python version is 3.11, and CUDA version is 12.1. The all datasets are randomly separated into training, validating, and testing sets following a ratio of 7:1:2. We set batch size to $64$ for all the training. Following MAE, we use a mask ratio of 75% for the pretraining. The pretraining epoch is 100 for 4 pathology image datasets. The implementation details are provided in A.1.

### 4.2 RESULTS AND DISCUSSION

We have carried out an extensive series of experiments, segmented into two principal components. The initial phase entailed a comprehensive assessment of the classification efficacy of our Masked Mamba model, juxtaposed against a selection of existing state-of-the-art (SOTA) models, across four

Table 2: Performance of our Masked Mamba trained with 4 Pathology image datasets.

| Classification network | LaC | | NCT | | PBC | | TCGA COAD | |
|---|---|---|---|---|---|---|---|---|
| | Acc(%) | F1(%) | Acc(%) | F1(%) | Acc(%) | F1(%) | Acc(%) | F1(%) |
| ResNet50 | 89.81 | 87.54 | 97.99 | 97.63 | 95.92 | 95.56 | 66.71 | 61.93 |
| ResNet101 | 90.18 | 89.05 | 98.92 | 98.65 | 96.14 | 95.07 | 67.64 | 69.73 |
| EfficientNet-b5 | 90.00 | 88.63 | 98.89 | 98.55 | 96.77 | 95.73 | 67.88 | 67.04 |
| MobileNet | 89.77 | 87.49 | 98.87 | 97.77 | 95.00 | 93.95 | 60.44 | 57.80 |
| ViT-B | 92.11 | 89.92 | 97.63 | 96.39 | 96.84 | 95.14 | 73.18 | 73.91 |
| Swin-Transformer-S | 93.40 | 91.96 | 97.20 | 97.00 | 96.88 | 97.13 | 77.83 | 76.00 |
| Swin-Transformer-B | 93.61 | 92.03 | 97.57 | 97.33 | 96.93 | 97.59 | 77.97 | 76.31 |
| VMamba-B | 92.13 | 90.40 | 91.57 | 90.80 | 85.33 | 87.19 | 73.08 | 77.17 |
| ViM-B | 90.57 | 89.35 | 90.00 | 90.07 | 83.73 | 86.55 | 71.90 | 76.00 |
| MAE | 98.60 | 96.31 | 98.99 | 98.71 | 98.05 | 98.83 | 87.90 | 89.17 |
| Masked Mamba | **99.84** | **98.47** | **99.46** | **99.38** | **99.17** | **99.54** | **90.18** | **91.89** |

diverse general pathology datasets. Subsequently, the second phase delved into a detailed comparative analysis, focusing on the distinct aspects of block structure and patch merging methodologies.

### 4.2.1 MASKED MAMBA EVALUATION

We employed four publicly accessible datasets of pathological imagery to substantiate the efficacy of Masked Visual Meta-learning (ViM) through a comparative analysis against nine contemporary state-of-the-art (SOTA) algorithms as enumerated in the comparative Table 2. Initially, within the framework of supervised classification paradigms, we designated ResNet50, ResNet101, EfficientNet-b5, MobileNet, ViT-B, Swin-Transformer-S, VMamba-B, and ViM-B as the benchmarks. To uphold equitable conditions, our experimental protocols incorporated pre-training and fine-tuning stages, with rigorous adherence to dataset uniformity. The empirical findings revealed that, within the purview of the LaC, PBC, and TCGA COAD datasets, the supervised classification schema Swin Transformer manifested the most better classification accuracy. Conversely, within the NCT dataset, the preeminent supervised classifier was identified as ResNet101. Masked Mamba demonstrated superior classification outcomes in comparison to the aforementioned supervised classifiers across all four pathological imagery datasets. Subsequently, Masked Mamba realized a marked enhancement in accuracy over the unsupervised MAE algorithm on the LaC, NCT, PBC, and TCGA COAD datasets by increments of 1.24%, 0.47%, 1.12%, and 2.28% respectively. The F1 scores correspondingly eclipsed those of MAE by margins of 2.16%, 0.67%, 0.71%, and 2.72% respectively.

MAE employs a high percentage of masks (typically 75%), which means that the model needs to learn from less visible information and predict a large amount of missing information. It can be seen from the data in the table that the classification results in LaC are better than other data sets. This suggests that this strategy can be more effective in learning the global and local features of high-resolution pathology images.

### 4.2.2 THE EFFECT OF DM BLOCK

In order to substantiate the efficacy of the DM Block, a series of comparative experiments were devised, specifically targeting the encoder's blocks. As delineated in Table 3, our novel DM integrated with a masking strategy was juxtaposed against the ViT-B and Mamba-B on four publicly accessible pathology datasets. The empirical data presented within the table demonstrate that the DM block possesses a definitive superiority across the LaC, NCT, and PBC datasets. Compared with traditional MAE, the accuracy of DM in LaC,NCT and PBC datasets is 0.69%, 0.48% and 0.95% higher, respectively. The sole instance where DM underperformed relative to ViT-B was on the TCGA COAD dataset, with a marginal decrease in accuracy of 0.74%. Consequently, aiming to augment the performance of our algorithm, we introduce a multi-scale patch merging strategy.

### 4.2.3 THE EFFECT OF LPPE

To ascertain the efficacy of LPPE, a series of comparative experiments were executed, integrating a variety of patch merging strategies with our DM block. These included the linear projection

Table 3: Performance of our DM Block trained with 4 Pathology image datasets.

| Method | Encoder | LaC | | NCT | | PBC | | TCGA COAD | |
|---|---|---|---|---|---|---|---|---|---|
| | | Acc(%) | F1(%) | Acc(%) | F1(%) | Acc(%) | F1(%) | Acc(%) | F1(%) |
| MAE | ViT-B | 98.60 | 96.31 | 98.99 | 98.71 | 98.05 | 98.83 | **87.90** | **89.17** |
| | Mamba | 98.55 | 96.25 | 98.87 | 98.47 | 97.76 | 97.47 | 87.04 | 88.16 |
| Masked Mamba (Linear) | DM | **99.15** | **97.16** | 98.98 | 98.86 | 98.22 | 98.17 | 87.16 | 88.55 |

Table 4: Performance of LPPE trained with 4 Pathology image datasets.

| Patch Merging | LaC | | NCT | | PBC | | TCGA COAD | |
|---|---|---|---|---|---|---|---|---|
| | Acc(%) | F1(%) | Acc(%) | F1(%) | Acc(%) | F1(%) | Acc(%) | F1(%) |
| Linear | 99.15 | 97.16 | 98.98 | 98.86 | 98.22 | 98.17 | 87.16 | 88.55 |
| Patch Merging | 99.68 | 98.02 | 99.31 | 99.02 | 98.98 | 99.11 | 87.25 | 89.93 |
| LPU | 99.41 | 97.20 | 99.22 | 98.90 | 98.70 | 99.04 | 87.47 | 90.15 |
| LPPE | **99.84** | **98.47** | **99.46** | **99.38** | **99.17** | **99.54** | **90.18** | **91.89** |

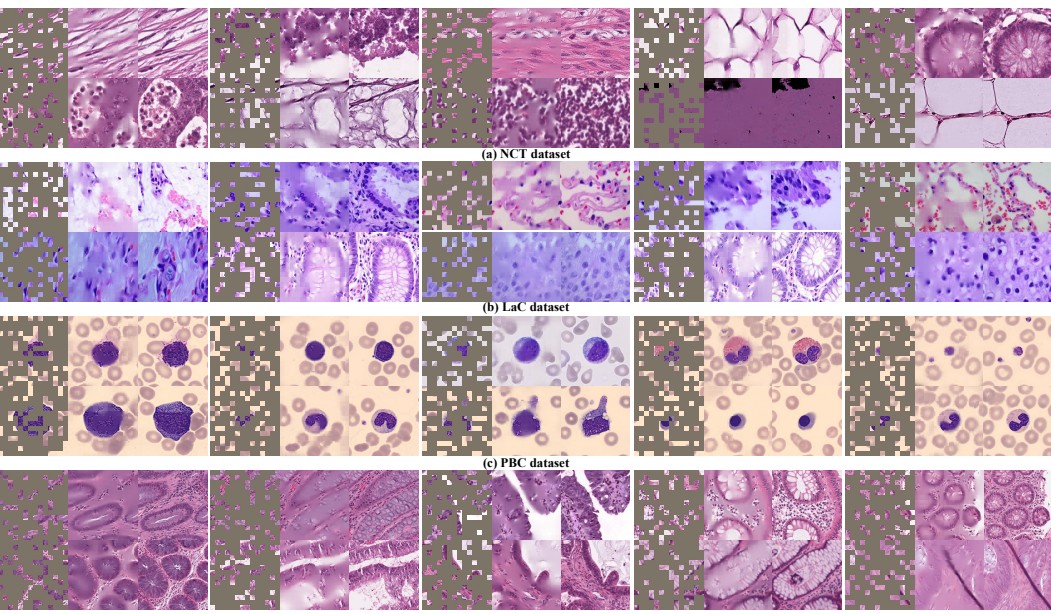

(a) NCT dataset

(b) LaC dataset

(c) PBC dataset

(d) TCGA COAD dataset

Figure 3: Uncurated random samples on publicly accessible pathology validation images. For each triplet, we show the masked image (left), our Masked Mamba reconstruction (middle), and the ground-truth (right). The images in rows 1 and 2 are from the NCT dataset, rows 3 and 4 are from the LaC dataset, rows 5 and 6 are from the PBC dataset, and rows 7 and 8 are from the TCGA COAD dataset. The masking ratio is 75%.

mechanisms present within ViT models, the sliding window downsampling technique from Swin Transformers, and our innovative LPPE approach. The empirical findings, as illustrated in Table 4, evidence the exceptional performance of LPPE on four publicly accessible pathology image datasets. Specifically, within the LaC dataset, LPPE realized a classification accuracy that surpassed the linear operations and window downsampling methods by 0.69% and 0.16%, respectively. For the NCT dataset, the respective improvements over linear operations and window downsampling were 0.48% and 0.15%. In the PBC dataset, the classification accuracy enhancements were noted to be 0.95% and 0.19%, respectively. Furthermore, within the TCGA COAD dataset, the LPPE strategy accomplished an accuracy of 90.18%. Although LPU position encoding performs well on TCGA COAD dataset, Acc and F1 of our LPPE are 2.71$ and 1.74% higher than LPU. Through the experimental data, it can be seen that LPPE shows stable and good performance on the above four histopathology datasets.

## 5 CONCLUSION

In this work, we introduced an innovative unsupervised classification algorithm tailored for the unique challenges of pathological image analysis. Central to this algorithm is the mitigation of dependency on extensively annotated datasets, which are often scarce and labor-intensive to produce.

Pathological image classification is inherently complex due to the intricate visual similarities that can exist across various cell types and tissues. This complexity is compounded by the necessity to focus on specific local features, such as the intricacies of cell nuclei morphology or the cytoplasm's distribution. To address these issues, we developed an advanced LPPE module for the encoding process, specifically designed to enhance feature extraction from critical local areas. Complementing this, we incorporated the DM module within the encoder to augment the classifier's capacity for assimilating global contextual information and mastering long-range spatial dependencies. This dual-module approach effectively reduces the over-reliance on particular pathological regions and accommodates the inherent variability in staining processes.

The robustness of our algorithm was rigorously evaluated through classification experiments on four diverse pathological datasets. The results were clear: our algorithm not only holds its own but outperforms current state-of-the-art methods, demonstrating its potential as a tool for advancing pathological analysis. Looking ahead, the versatility of our proposed algorithm opens up promising avenues for application in clinical tasks, where it has the potential to facilitate more accurate diagnostics and contribute to the broader field of medical image analysis. This research marks an important step forward in the quest for more effective and efficient pathological image classification, and we are optimistic about its future applications and continued development.

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

# A APPENDIX

## A.1 IMPLEMENTATION DETAILS

The comprehensive experimental settings for the pre-training and downstream tasks are provided in Table 5a and Table 5b, respectively.

Table 5: Parameter setting

| Config | Value |
|---|---|
| Optimizer | AdamW Loshchilov & Hutter (2017) |
| Base learning rate | 5e-5 |
| Weight decay | 0.05 |
| Optimizer momentum | $\beta_1$, $\beta_2$ = 0.9, 0.95 |
| Batch size | 64 |
| Learning rate schedule | cosine decay Loshchilov & Hutter (2016) |
| Warmup epochs Goyal et al. (2017) | 10 |
| Augmentation | RandomResizedCrop |

(a) Pretraining setting.

| Config | Value |
|---|---|
| Optimizer | AdamW |
| Base learning rate | 1e-3 |
| Weight decay | 0.05 |
| Optimizer momentum | $\beta_1$, $\beta_2$ = 0.9, 0.999 |
| Layer-wise lr decay | 0.75 |
| Batch size | 64 |
| Learning rate schedule | cosine decay |
| Warmup epochs | 5 |
| Augmentation | RandAug (9, 0.5) Cubuk et al. (2020) |
| Label smoothing | 0.1 |
| Drop path | 0.1 |

(b) Classification test transfer setting.

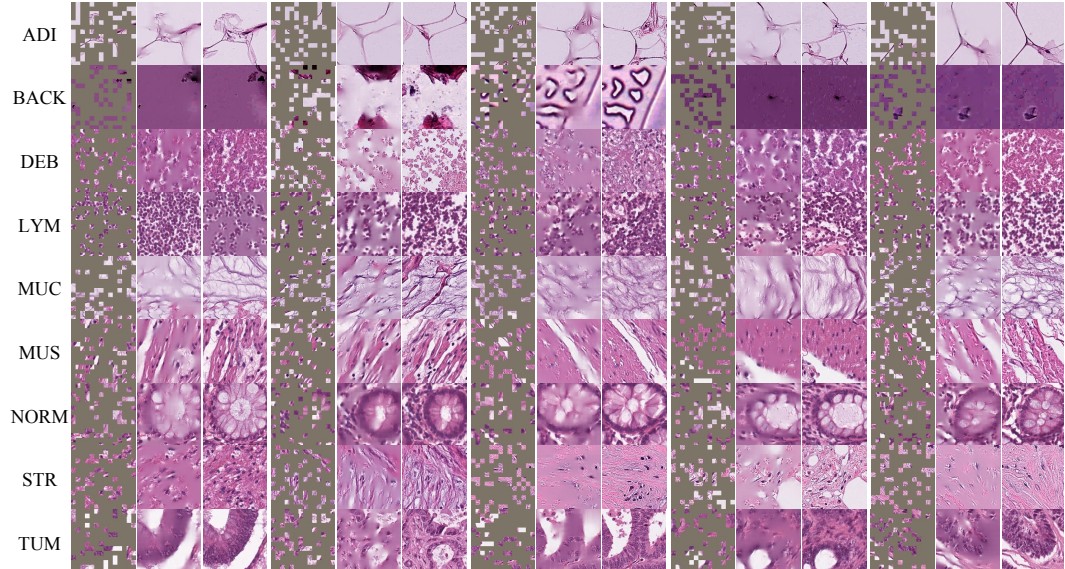

Figure 4: Uncurated random samples on NCT. For each triplet, we show the masked image (left), our Masked Mamba reconstruction (middle), and the ground-truth (right). The masking ratio is 75%.

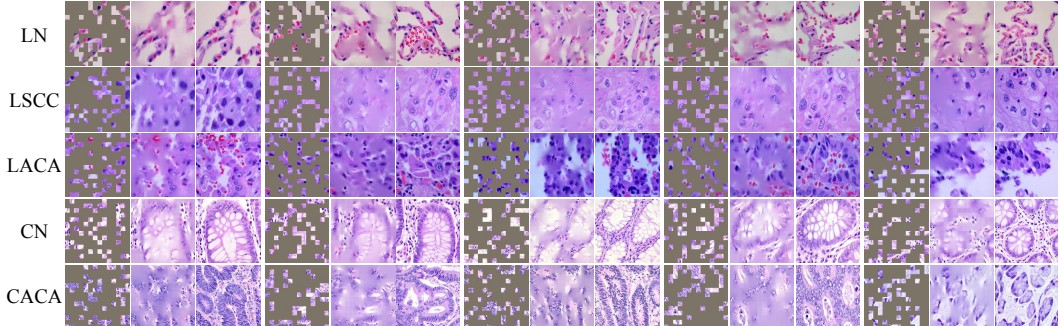

Figure 5: Uncurated random samples on LaC. For each triplet, we show the masked image (left), our Masked Mamba reconstruction (middle), and the ground-truth (right). The masking ratio is 75%.

### A.2 MORE VISUALIZATION RESULTS ON PATHOLOGY IMAGES

The NCT dataset consists of 9 distinct classes, which are as follows: Adipose (ADI), Background (BACK), Debris (DEB), Lymphocytes (LYM), Mucus (MUC), Smooth Muscle (MUS), Normal Colon Mucosa (NORM), Cancer-Associated Stroma (STR), Colorectal Adenocarcinoma Epithelium (TUM). This dataset is a collection of 100,000 non-overlapping image patches from hematoxylin and eosin stained histological images of human colorectal cancer (CRC) and normal tissue. The visualization for the above 9 classes is shown in Figure 4.

The LaC dataset contains 25,000 athological images with 5 classes, which are as follows: lung tissue (LN), lung adenocarcinomas (LACA), lung squamous cell carcinomas (LSCC), colon tissue (CN), colon adenocarcinomas (CACA). All images are $768 \times 768$ pixels in size and are in jpeg file format. There are 5 classes in the dataset, each with 5,000 images. The visualization for the above 5 classes is shown in Figure 5.

The PBC dataset consists of 17,092 images. These images are further organized into the following 8 groups: neutrophils (NE), eosinophils (EO), basophils (BA), lymphocytes (LY), monocytes (MO), immature granulocytes (IG), erythroblasts (ERB), and platelets (PL). Each image is $360 \times 363$ pixels in size and is in JPG format, annotated by expert clinical pathologists. The visualization for the above 8 classes is shown in Figure 6.

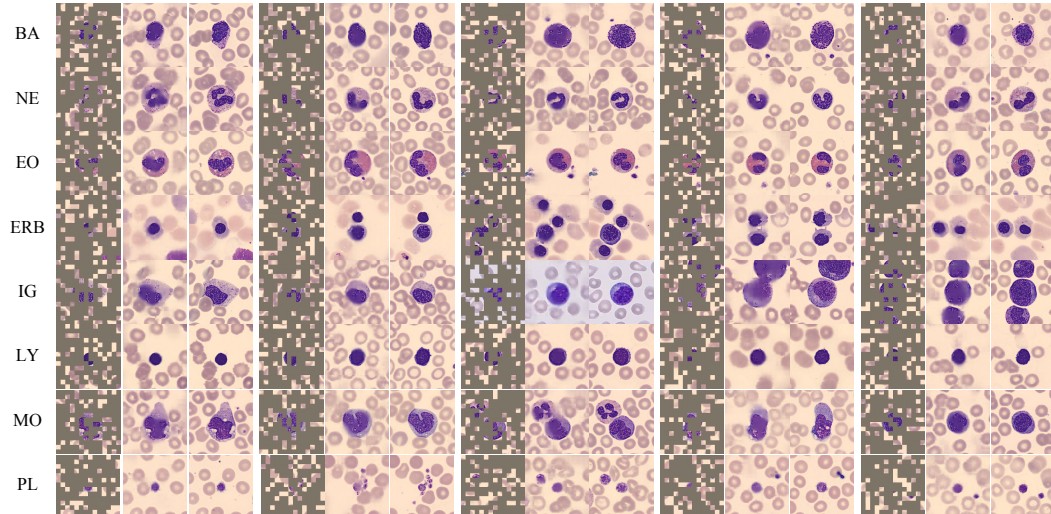

Figure 6: Uncurated random samples on PBC. For each triplet, we show the masked image (left), our Masked Mamba reconstruction (middle), and the ground-truth (right). The masking ratio is 75%.

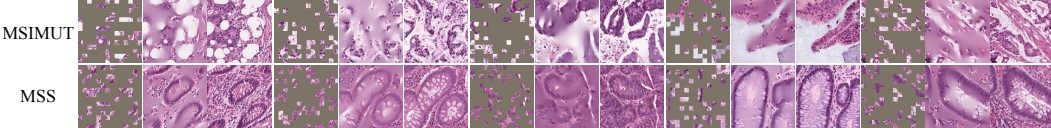

Figure 7: Uncurated random samples on TCGA COAD. For each triplet, we show the masked image (left), our Masked Mamba reconstruction (middle), and the ground-truth (right). The masking ratio is 75%.

The TCGA COAD dataset contains 192312 unique image patches derived from histological images of colorectal cancer and gastric cancer patients in the TCGA cohort. The dataset encompasses two categories: "MSS" (microsatellite stable) and "MSI" (microsatellite unstable or highly mutated). It has been utilized for the automatic detection of tumors. The pixel dimensions of the images within this dataset are $224 \times 224$ pixels. The visualization for the above 2 classes is shown in Figure 7.

