# OpenReview forum: "Masked Mamba: An Efficient Self-Supervised Framework for Pathological Image Classification"
_ICLR.cc/2025/Conference — Submitted to ICLR 2025_

### Official Review · Reviewer_F4TG · 2024-10-31

**Soundness:** 3
**Presentation:** 2
**Contribution:** 2
**Rating:** 3
**Confidence:** 5

**Summary:**

This paper proposes Masked Mamba, an efficient model for pathological image classification. To adapt Mamba for pathological images, it introduce a patch ghosting module to capture multi-scale features and employ masked autoencoders to extract robust feature representations. Experimental results demonstrate that Masked Mamba achieves state-of-the-art performance.

**Strengths:**

1.	The proposed patch ghosting module effectively enhances locality with multi-scale information, making it highly practical.
2.	Extending Mamba from 1D sequence modeling to 2D image modeling is a valuable and well-motivated approach.

**Weaknesses:**

1.	The combination of Vision-Mamba and masked autoencoders shows limited novelty, as neither component is original to the authors.
2.	The patch ghosting module should be compared to the commonly used Local Perception Unit described in [1].
3.	Although the paper claims that Masked Mamba is efficient, it lacks a table comparing parameters or FLOPs to substantiate this claim.
4.	This paper resizes all pathological images to 224×224, which fails to convincingly demonstrate that Mamba is applicable to high-resolution pathological images.
5.	The writing and expression could be improved, as the contributions in the abstract and introduction are inconsistent.
[1] Guo J, Han K, Wu H, et al. Cmt: Convolutional neural networks meet vision transformers[C]//Proceedings of the IEEE/CVF conference on computer vision and pattern recognition. 2022: 12175-12185.

**Questions:**

No question

---

> ### Author Response · Authors · 2024-11-19
>
> We sincerely appreciate the constructive suggestions from reviewers.
> First, instead of using the traditional Vision-Mamba, we have redesigned it. We use depthwise separable convolutions and regular convolutions instead of causal convolutions to boost the order and spatial information in the feature representation. Since patch embedding is replaced by patch ghost designed by us, it is not a pure fusion when using MAE.
>
> Secondly, The Local Perception Unit in [1] can be expressed as LPU(X) = DWConv(X)+X. In contrast, our Patch Ghosting can be represented as Ghost(X) = Conv(DWConv(Conv(X))+Conv(X)).
>
> We'll add Params and FLOPs later. The models we use often require input images to be of a consistent size (e.g. ViT, Swin Transformer, VMamba, etc.), but the images in the dataset we use are not 224.
>
> Our revised abstract is as follows: "Extracting visual representations is a crucial challenge in the domain of computational histopathology. Considering the powerful strength of deep learning algorithms and the dearth of annotated samples, self-supervised learning presents itself as a compelling strategy to extract effective visual representations from unlabeled histopathology images. Although some self-supervised learning methods have been specifically proposed for histopathology image classification, most of them have certain drawbacks that may affect the functionality or representation capacity. In this work, we propose Masked Mamba, a novel self-supervised visual representation learning method tailored for histopathology images that can adequately extract local-global features. The proposed method consists of two stages: local perception positional encoding (LPPE) and directional Mamba vision backbone (DM). In addition, we use masked autoencoder (MAE) pretraining to unleashing directional Mamba vision backbone's potential. Masked Mamba makes good use of domain-specific knowledge and requires no side information, which means good rationality and versatility. Experimental results demonstrate the effectiveness and robustness of masked Mamba on common histopathology classification tasks. Furthermore, ablation studies prove that the local perception positional encoding and directional Mamba vision backbone in masked Mamba can complement and enhance each other."
> At the same time we modify the presentation of the contribution of our work in the introduction.

---

### Official Review · Reviewer_AuG2 · 2024-11-03

**Soundness:** 2
**Presentation:** 2
**Contribution:** 2
**Rating:** 3
**Confidence:** 4

**Summary:**

The paper proposes a self supervised model for pathology image tasks that is based on the Mamba model and Masked auto encoder. The propsoed method modifies the convolution operation in Mamba and the patch merging procedure. Additionally it is trained with 75% masked input and learns to reconstruct the input images. The model is evaluated on 4 pathology datasets on image classification tasks and shows improved performance compared to various baselines.

**Strengths:**

- The paper proposes modifications to the Mamba model to make it more suitable for pathology image analysis.
- The experimental results show improved performance compared to various baselines.
- The proposed method tries to take advantage of the feature robustness provided by Masked auto encoders to enhance the training of Mamba model.

**Weaknesses:**

- Using SSM is listed as a contribution even though it is already an integral part of the Mamba model adapted in the paper.

- Also from the contributions: "By leveraging a blend of deep separable and regular convolutions as alternatives to traditional causal convolutions, our
approach reinvents the extraction and sequentialization of spatial features,". This is an over statement. A combination of separable and regular convolutions have been used in previous models, Inception to name one.

- The intuition behind the patch ghosting operation is not clear. I'm not sure where the name comes from and why it is better than other patch merging operations. Even the ablation studies don't show significant improvement in 3 out of 4 tasks.

- The model proposed even though sounds general is only evaluated and targeted towards pathology image classification tasks. No segmentation of WSI classification, and no other types of medical or natural image datasets.

- Datasets description is lacking:
	- The reference for the dataset TCGA COAD: Couture (2022) is a review paper and not a dataset paper.
	- There is no mention of the datasets tasks, their labels, and class-wise statistics.
	- It is not clear whether there is a data split that is published with the datasets or the authors split the data.
	- If the split was done by the authors, there is no mention of how the splitting performed other than the ratio and there no cross validation evaluation.
	- It is mentioned that the patches are resized to 224 by 224 but the original magnification of the datasets is not mentioned.

- The following statement needs clarification: "the resolution in pathological images is often influenced by staining and sectioning"

- The evaluation does not include more current models that have shown good performance on pathology images, such as pathology foundational models, CTransPath, PLIP,

- The improvement in performance is mostly fractional. It is not clear how signficant are the results.

- In Equation 6, the second line, it is not clear what double arrows mean.

- In line 288, SSD was not mentioned before.

- Typo: line 161: The 1 is transformed into a discrete function
- Typo: eq 11: incorrect sign (1+yn)

**Questions:**

Please refer to the weaknesses.

---

> ### Author Response · Authors · 2024-11-20
>
> We sincerely appreciate the constructive suggestions from reviewers.
> 1. Tackling the critical dependency on localized regions and the inherent heterogeneity of staining procedures in pathological images, we redesigned the Mamba module for the pathology image classification task.
>
> 2. We revised the statements in the manuscript: "Specifically, we replace causal convolutions with depthwise separable convolutions and standard convolutions to better capture global context."
>
> 3. The absolute position encoding used in previous Mamba, originally designed to exploit the order of tokens, ignores local relations and structural information inside the patches. Therefore, we redesigned an encoding to extract local information. "Linear" in Table 4 represents the original way. To illustrate the effectiveness of our "Patch Ghosting", we also compare the encoding strategy (Patch Merging) in Swin Transformer. Experimental results our "Patch Ghosting" works best. So I don't understand the meaning of the reviewer's "Even the ablation studies don't show significant improvement in 3 out of 4 tasks."
>
> 4. We chose pathological images because of their characteristics of high similarity. Therefore, pathological section images with highly similar features were selected in our experiments.
>
> 5. We really appreciate your suggestions for our dataset. The first thing we need to mention is that the datasets we use are all pathology datasets that have been preprocessed. TCGA COAD is the dataset obtained after dividing the TCGA dataset into patches. The four pathological image data sets involved in this paper are all divided, and these data sets are not divided by our team. If the resolution of the original WSI image is required, we can add the description in the appendix.
>
> 6. We modified this to "The quality of staining and sectioning directly affects the sharpness of the image and the visibility of details. Uneven staining or sections that are too thick/thin can lead to image quality degradation and affect the pathologist's diagnosis and analysis."
>
> 7. The performance of CTransPath on the NCT dataset is 96.52 for Acc and 94.82 for F1. Our result is 99.46 and 99.30. PLIP was a pathology domain-specific vision–language pretrained model using image–text pairs from Twitter.  Obviously our study is not aimed at multimodality. Thank you very much for your comments. I think we will add a lot of experimental data of self-supervised models to illustrate the advantages of our algorithm in the future.
>
> 8. We disagree with you that the improvement in performance is insignificant. Perhaps we will increase the performance of some of the models already proposed. But from our current experiments, it can be seen that the classification results of our algorithm are the best.
>
> 9. Double arrows means forward and backward propagation. Thank you for your advice on the details.
>
> 10. This is a typo. It's essentially an SSM.
>
> 11. The Eq. 1 is transformed into a discrete function.
>
> 12. This is a typo. It's essentially "1-yn".

---

> > ### Comment · Reviewer_AuG2 · 2024-12-02
> >
> > I appreciate the authors responses. However, similar to other reviewers I find the novelty and contribution are limited. I maintain my original score.

---

### Official Review · Reviewer_adhC · 2024-11-04

**Soundness:** 2
**Presentation:** 2
**Contribution:** 2
**Rating:** 3
**Confidence:** 5

**Summary:**

The authors proposed a new, self-supervised framework designed to improve the classification of pathological images. The framework includes a unique Patch Ghosting module that captures local image features effectively, and a MixedMamba Block to enhance the model's understanding of global and long-range dependencies. This proposed design helps overcome issues related to limited high-quality annotated data and variability in sample staining.

**Strengths:**

The authors proposed a newer feature representation learning method using Mamba.

**Weaknesses:**

1.	The quality of the paper is poor. The equations do not aid the audience in understanding the work and lack significant details.
2.	The title claims that the method is unsupervised. However, training the classification head still requires labels, making the claim of unsupervised classification inaccurate. The Mamba-based unsupervised autoencoder only extracts feature embeddings. Therefore, the work should be described as unsupervised feature representation learning with a supervised classification method.

**Questions:**

1. In line 161, “The 1 is transformed…,” what does '1' refer to?
2. In equation 1, what do A, B, and C represent? The authors only provided the dimensions of these matrices but did not explain their significance.
3. The proposed model's accuracy and F1 scores are exceptionally high for the LaC dataset compared to other models. The LaC dataset features are known to be extremely difficult to distinguish. Please address how the proposed method effectively learned the embedding space and extracted features that can be utilized efficiently.

---

> ### Author Response · Authors · 2024-11-19
>
> We sincerely appreciate the constructive suggestions from reviewers.
> First of all, we will further optimize the writing quality of our paper, and according to your suggestion, we think its title can be changed to "EMP: Efficient Mamba Backbone's Potential with Masked Autoencoders pretraining for Pathological Image Classification"
>
> In response to your question, we have made The following modifications to the paper:
> 1. "The 1 is transformed... "The Eq. 1 is transformed... "The eq. 1 is transformed... ,"
> 2. we explain A, B and C in the manuscript. It is shown as "where A represents the state matrix, B and C denote the projection parameters."
> 3. MAE employs a high percentage of masks (typically 75%), which means that the model needs to learn from less visible information and predict a large amount of missing information. It can be seen from the data in the table that the classification results in LaC are better than other data sets. This suggests that this strategy can be more effective in learning the global and local features of high-resolution pathology images.
> 4. To illustrate that our algorithm indeed effectively learns the embedding space and extracts features, we include the results of visualization in our experiments. You can refer to Figure 3. For each triplet, we show the masked image (left), our Masked Mamba reconstruction (middle),  and the ground-truth (right). The images in rows 1 and 2 are from the NCT dataset, rows 3 and 4 are from
> the LaC dataset, rows 5 and 6 are from the PBC dataset,  and rows 7 and 8 are from the TCGA COAD dataset. The masking ratio is 75%.

---

> > ### Comment · Reviewer_adhC · 2024-11-27
> >
> > Thank for addressed the few things I pointed out. However, the overall quality of the paper is not improved. The novel contribution is limited. I agree with other reviewer and aligned my score with other reviewers.

---

### Official Review · Reviewer_x8Dw · 2024-11-11

**Soundness:** 3
**Presentation:** 2
**Contribution:** 3
**Rating:** 3
**Confidence:** 2

**Summary:**

The authors have built an image encoder self-supervised learning training tech for histopathological patches from whole slide images.  The technique uses principles of masked image encodinf (i.e. masked autoencoder) and principle for training recurrent neural networks (i.e. mamba) to build masked mamba.  The authors also built a novel model for subpatch merging called patch ghosting that is incorporated in the the masked mamba training archetecture.  The authors benchmark their technique on several patch level classfication tasks against standard architectures including MAE with favorable performance.  The authors also show that within their architecture patched ghosting outperforms ViT and Swin transformer.

**Strengths:**

The concept is novel.  We are interested in any and all SSL techniques that improve upon others for feature extraction of downstream tasks.

The model does show modest improvement on chosen benchmarks relative to MAE.

The masked mamba approach, relative to ViT and Swin Transformer is also a novel approach that could be explored in more detail.

**Weaknesses:**

I believe that a some portions  of the text (not the entire paper) was written with an LLM.  It makes the paper read a little hyperbolic with unneeded adjectives and superlatives.  A sentence like "... manifested the most exemplary classification proficiency." is not likely something someone would write.  Although I cannot certain.

Our experience is that masking strategies have yielded poor results for feature extraction for downstream tasks.  Not benchmarking with more successful techniques like DINO doesn't make sense to me. If authors can show performance relative to DINO that would greatly strengthen paper.

Generally speaking feature extraction encoders are most useful for allowing whole slide image classification tasks.  Any experiments showing performance on a useful whole slide image task would enhance this work.

**Questions:**

Given that the model is not segmenting cells, how can you claim "Therefore, our Patch Ghosting generates feature maps that encourage the model to capture diverse feature representations of similar cells." I don't see experiments that support such a claim.

---

> ### Author Response · Authors · 2024-11-19
>
> We sincerely appreciate the constructive suggestions from reviewers.
> First, we need to state that our paper was not written using LLM, but rather used LLM for language polishing. In the article "... manifested the most exemplary classification proficiency." has been modified to "... manifested the most better classification accuracy."
>
> Second, I need to illustrate that DINO is trained by self-distillation, which can improve the generalization ability of the model but may require more computational resources to train the teacher and student networks. But this is different from the efficient self-supervised pre-trained models we propose. Because we are using a masked image model in a self-supervised technique, it is different from masking strategies. Masked Autoencoders is a self-supervised learning method that randomly masks some patches in an image and then uses the autoencoder to predict these masked parts. However, masking strategies refer to the strategies that partially mask or mask the input data in self-supervised learning. MAE uses a lightweight architecture in the decoder stage, and the amount of computation for each token is only less than 10% relative to the decoder. This design makes it possible that the consumption of computational resources does not increase significantly even if the full number of tokens is used as input in the decoding stage. In addition, the autoencoder structure of MAE allows the model to focus on learning the features of the unmasked parts, which helps to reduce the computational burden. If you still think the comparison is needed, please let us know in time, and we will supplement it in the subsequent experiments.
>
> Finally, since the WSI is usually a large resolution image. It is usually necessary to preprocess the whole WSI, which includes dividing the image patches. Therefore, the pathology images we used were partitioned.
>
> Our explanation for your question is as follows: traditional Patch embedding usually only has a simple linear layer, so it will result in a large number of similar feature maps, most of which are redundant. Therefore, we adopt a DWConv and residual structure aiming to provide more information for the network.

---

### Meta-Review · Area_Chair_K2by · 2024-12-16

**Metareview:**

This paper proposes a State-space-model-based model (dubbed Masked Mamba) for unsupervised learning of pathology images. The designed modules aim to provide global-local representation learning capabilities, including local perception positional encoding (LPPE) and directional Mamba vision backbone (DM). Experimental results show improvement over baselines on several datasets.

This paper received 4x Reject ratings from reviewers. The main concerns raised by the reviewers centered around limited novelty (integration of Mamba and Masked Autoencoder has already been done by previous works) and poor quality of the paper (writing and expression could be improved). Although the authors solved some of the concerns during the rebuttal, the main concerns remain unaddressed. For example, reviewer F4TG stated, "Although the paper claims that Masked Mamba is efficient, it lacks a table comparing parameters or FLOPs to substantiate this claim." yet the authors did not include additional experiments to support their statement. Meanwhile, AC finds that the baseline models for comparison are mostly from the natural image domain, lacking specific baselines from the pathological domain.

Given the consensus of reviewers, rejection is recommended.

**Additional Comments On Reviewer Discussion:**

Reviewers admit this promising topic, yet their main concerns centered around this paper's limited novelty and unprofessional writing. During the rebuttal, the authors mainly addressed writing-related concerns. However, the experiment-related concerns are not fully addressed, which cannot strongly support the novelty of this paper.

---

### Decision · Program_Chairs · 2025-01-22

Reject